# The changes of willow biomass characteristics during the composting process and their phytotoxicity effect on *Sinapis alba* L.

Józef Sowiński[1]*, Anna Jama-Rodzeńska[1], Peliyagodage Chathura Dineth Perera[1,2], Elżbieta Jamroz[3], Jakub Bekier[3]

**1** Institute of Agroecology and Plant Production, Wroclaw University of Environmental and Life Sciences, Wrocław, Poland, **2** Faculty of Agriculture, Department of Agricultural Biology, University of Ruhuna, Mapalana, Kamburupitiya, Sri Lanka, **3** Institute of Soil Sciences Plant Nutrition and Environmental Protection, Wroclaw University of Environmental and Life Sciences, Wrocław, Poland

\* jozef.sowinski@upwr.edu.pl

**Data Availability Statement:** All relevant data are within the paper and its Supporting Information files.

## Abstract

This study evaluated in 2019–2021 the use of willow chips for compost production and its effect on *Sinapis alba* L. germination index and seedling growth. Peatlands and peat are of very important economic but above all environmental significance. The conservation of peatland resources is one of the most crucial future challenges. Composts and other forms of lignin-cellulosic biomass are potentially the best renewable alternative to peat in its economic use. Composted lignin-cellulosic biomass can replace peat and be used as a substrate for vegetable transplant production. The impact of modifying the willow lignin-cellulosic biomass composting process has not been well analysed. A compost experiment with willow biomass was conducted to study its effect on selected compost indexes (particle size structure in %, bulk density (kg m$^{-3}$), and total nitrogen content). The quality assessment of the willow composts was determined after six months of composting process based on the N content and morphological characteristics of tested plant in vegetative chamber. *Sinapis alba* L. was germinated on a water extract made from willow compost using the following additives to willow biomasses: W0—without additives, WN—with the addition of nitrogen, WF—with the addition of mycelium, WNF—with the addition of nitrogen and mycelium. During the composting process, samples were taken after each mixing of the biomass pile to assess their maturity through the use of a bioassay. Willow biomass did not have a negative effect on biological evaluation parameters, and in some indicators, such as the length of embryonic roots in the VI period of the measurements, it was stimulating (61–84% longer in W0 and WF than in the control). The addition of nitrogen during the composting process, especially in the initial composting period, had a strong inhibitory effect.

## 1. Introduction

Currently, one of the problems affecting the environment is the high level of carbon dioxide emissions. The most important task faced by humanity today is to reduce greenhouse gas

**Funding:** The research is co-financed under the Leading Research Groups support project from the subsidy increased for the period 2020–2025 in the amount of 2% of the subsidy referred to Art. 387 (3) of the Law of 20 July 2018 on Higher Education and Science, obtained in 2019. The APC/BPC is co-financed by Wroclaw University of Environmental and Life Sciences.

**Competing interests:** The authors have declared that no competing interests exist.

(GHG) emissions. One of the ways to reduce the exploitation of stored carbon resources in the Earth's crust its sequestration in the top layer of the Earth is increase of lignocellulosic biomass use in circular bioeconomy [1].

Peatland is one of most globally important stored carbon resources and the bound carbon stocks (in terms of quantity on a global scale) exceed those bound by tropical forests (500–700 billion tons of C compared 360 billion tons of C, respectively) [2].

Horticulture, agriculture and forestry, make extensive use of peatlands for various types of production. It is estimated that 10% of the GHG emitted into the atmosphere comes from exploited peatlands [3]. Quantis [4] points out that peat has the greatest impact on "climate change" used as a resources of horticulture substrate material. Joosten and Clarke [5] report that peat extraction for horticultural growing media is carried out on about 2000 km$^2$ of peatland. Peatland drainage leads to degradation and, in conjunction with climate warming, rising temperatures, and drier weather patterns, can lead to excessive mineralization and destabilization of peat C stores. In extreme cases fires may break out [2,6]. Therefore its use is limited because of its long-lasting effect and reduced nutrient delivery to crops.

Today we are encouraged to use alternative substrates in horticulture production aiming to almost complete replacement of peat. The main reason for this is the need to protect environment and the increasingly recognized desire for environmental sustainability, while maintaining a competitive horticultural industry [7]. Therefore, research is being carried out related to the search for other organic and mineral materials that could be used as a replacement or additive to horticultural substrates that limit the use of peat and incorporating new product into circular processes [8–10]. Lignocellulosic feedstocks can be divided into three categories: i.e. forest residues, agricultural residues and herbaceous and woody crops cultivated as a Short Rotation Coppice (SRC). High biomass yielding crops, particularly perennial grasses (miscanthus, switchgrass, prairie grass, and short rotation forest species such as eucalyptus, poplar, and willow) are considered in this direction [11]. Willow for biomass can be cultivate on a range of environmental condition (including marginal land) up to 20 years with multiple harvests on three- to four-year cycles. Shredded willow wood biomass undergoes the biotransformation process of the lignocellulosic complex into stable, more complex humic substances [12]. Compost constitutes a source of macronutrients, being one of the best methods of utilization of various biomasses, including the difficult to utilize, to reuse it for agricultural, reclamation, or horticultural purposes as a horticulture media [13]. The application of compost allows carbon to be sequestered in the form of stable organic matter, increases, and keep soil organic carbon at stable levels, therefore, it is one of the key objectives in the priorities of the European Union's [14]. It is also known to improve the physical properties of excessively hydrated biomass by adding wood waste or sawdust to the composting process [15] or shredded wood waste or wood can be fully composted [12,16].

One of the most important factors that affect the quality of the final product is the proper selection of biomass for the composting process. Components of the biomass should have an appropriate carbon to nitrogen (C:N) ratio [17,18]. When the C:N ratio is too wide–over 40, the organic matter decomposition processes are inhibited. Conversely, in the case of very narrow C:N ratio, below 10, rapid organic matter decomposition and significant losses of nitrogen due to ammonia release are observed. It has a toxic effect on microorganisms, especially at high pH and temperature values as well as can lead to inhibition of the organic matter transformation [19]. The appropriate selection of components and the ensuring of optimal conditions during the composting process create the possibility of the compost use as valuable product.

Considering the consequences of the above-mentioned horticultural media, willow wood biomass can be considered as a good potential for an alternative to peat, and it has far less studied. The using of advantages according to the literature on nitrogen compounds and the

biological additive (fungal spores of *Peniophora gigantea*) during the compost production processes modify the decomposition process while desiring fertile horticultural media. The knowledge of the properties of composted willow chips with an additives and their relations occurring during the composting process allows for their use as a substrate with an appropriate additive in the future. The physical properties of compost materials change including willow chips hence undertaking research to evaluate this technology against willow chips and its further use. The beneficial effect of composted willow biomass as a horticultural substrate for growing tomato transplants and for growing lettuce at early development stages was confirmed in studies by Adamczewska Sowińska et al. [9], Bekier et al. [12].

We hypothesized that willow composting would change all the examined features of this material, as well as morphological traits and germination index [20]. This article presents the effect of different methods of composting willow chips on the change of selected parameters of compost during its production, as well as the quality of composted biomass assessed by the germination index with the use of the indicator plant *Sinapis alba* L. The impact of the water extracts made from the composting biomass of *Salix viminalis* L. on different biotransformation processes on the plant growth of the tested plant under controlled conditions was determined. Therefore, in present research N and N plus fungi have been proposed as additives to willow biomass with wide C:N ratio to examine its effect on N content and physical parameters. The studies comprehensively capture the effect of compost additives on the phytotoxicity of *Sinapis alba* L. in a germination test.

Peatlands with a high carbon accumulation is crucial land for global climate change and biodiversity protection. Presented research was carried out of searching other alternative organic materials that could be used as a horticultural media, limits peat utilisation and exploitation. Use of willow wood biomass for growing media is novel methods. Willow renewable biomass undergoes the **l**ignocellulosic complex biotransformation on seminatural composting process into humic substances. During the compost production processes cyclically biomass samples were taken for determine the bioassay stimulation of germination capacity or toxicity potential of the willow biomass.

## 2. Materials and methods

### 2.1 Sampling sites and experiment design

The willow biomass (*S. viminalis* L.) was harvested from SRC in Pawłowice, Wrocław (51° 11'N, 17°08'E). The plantation was established in 2003 and regularly harvested every three years for energy purposes. The last harvest was in winter 2015. For composting process harvesting took place in 2019 using a special chopper. The chops size ranged from 4 to 12 mm. The willow chips were immediately transported to Research and Didactic station Psary where the composting process started under semi natural conditions by tipping the chips onto the ground with limiting treatments during the composting process to the necessary minimum. The biomass of the willow plant was cut into chips uniform organic material with the dominant fraction in terms of <1.9 cm (46.3%), following fraction >1.9 cm (42.0%) and below 0.8 cm (11.7%) in average. The size of the willow's chips has impacts of the composting process. The willow chip size can be regulate by the speed of the feedrolls or the number of knives on the harvester drum. The willow chips were composted for a period of six months in four biomass piles with different additives according to semi-dynamic open pile system [21]. The willow chip compost prisms were formed on a horticultural mat. The prisms were 1.5 m wide, 1.3 m high and 5 m long. The volume of the prisms was approximately 5 m³. The prisms were successively irrigated and manually mixed at 4–5 week intervals. The different additives were added once one month after the start of composting. During composting processes, the

**Table 1. List of sampling dates along with the duration of the composting process.**

| Time from composting start | Date of sampling | Additive to willow compost | Abbreviation |
|---|---|---|---|
| 2 days | 23.03.19 | - | W0 |
| 33 days | 10.04.19 | - | W0 |
| 33 days | 10.04.19 | Fungi | WF |
| 33 days | 10.04.19 | Nitrogen in a form of $NH_4NO_3$ | WN |
| 33 days | 10.04.19 | Nitrogen in a form of $NH_4NO_3$ plus fungi | WNF |
| 64 days | 10.05.19 | - | W0 |
| 64 days | 10.05.19 | Fungi | WF |
| 64 days | 10.05. 19 | Nitrogen in a form of $NH_4NO_3$ | WN |
| 64 days | 10.05.19 | Nitrogen in a form of $NH_4NO_3$ plus fungi | WNF |
| 95 days | 5.06.19 | - | W0 |
| 95 days | 5.06.19 | Fungi | WF |
| 95 days | 5.06.19 | Nitrogen in a form of $NH_4NO_3$ | WN |
| 95 days | 5.06.19 | Nitrogen in a form of $NH_4NO_3$ plus fungi | WNF |
| 143 days | 17.07.19 | - | W0 |
| 143 days | 17.07.19 | Fungi | WF |
| 143 days | 17.07.19 | Nitrogen in a form of $NH_4NO_3$ | WN |
| 143 days | 17.07.19 | Nitrogen in a form of $NH_4NO_3$ plus fungi | WNF |
| 189 days | 15.09.19 | - | W0 |
| 189 days | 15.09.19 | Fungi | WF |
| 189 days | 15.09.19 | Nitrogen in a form of $NH_4NO_3$ | WN |
| 189 days | 15.09.19 | Nitrogen in a form of $NH_4NO_3$ plus fungi | WNF |

temperature and biomass was monitored and complete composting was established on temperature to surrounding outdoor temperature. In this experiment, the following properties of willow compost was performed: total nitrogen (%), bulk density (kg m$^{-3}$) and percentage of fraction particles in the compost (%).

The nitrogen addition (chemical) was used to change the wide C:N ratio and the biological addition (fungal spores) was used to accelerate the decomposition of lignin-cellulosic compounds of the biomass. During the period from March to the end of July, compost production occurred according to the processes listed below and was cyclically subjected to mixing as indicated by the dates below (Table 1). Additionally, samples were taken two months after the end of the composting process (September):

➢ Shredded willow chips (W0),

➢ Shredded willow chips plus nitrogen in the form of ammonium nitrate (34% N), added to change the C:N ratio (WN).

➢ Shredded willow chips with the addition of the wood decay fungus, *Peniophora gigantea* (WF).

➢ Shredded willow chips with the addition of wood decay fungi and nitrogen (WNF).

For technical reasons, the biomass structure and bulk density of the compost were not determined on date VI. Every month, the piles were mixed and immediately after this process (April, May, June, July) the samples were taken to determine percentages of individual biomass fractions, specific compost weight, bulk density, and total N content. Additionally, a sample for only the N content and the biological test was collected in September (after finishing the

composting process). Reagents used for the determination of total nitrogen were as follows: sulphuric acid 96% (pure for analysis)—CAS catalogue number: 7664-93-9, granulated sodium hydroxide (at a concentration of 30%, pure for analysis)—CAS catalogue number: 1310-73-2, hydrogen peroxide 30% (pure for analysis)—CAS catalogue number: 7722-84-1. The reagents originally came from the company: P.P.H. "STANLAB" Sp. z.o.o Lublin.

The following samples were taken into account from different dates to determine the germination capacity of the willow biomass, toxicity potential, and bioassay (Table 1).

Samples of composted willow biomass collected at the above dates were subjected to the following parameters.

The particle size distribution was assessed using special Pennsylvania sieves (with four different fractions: <0.8 cm, 08–1.9 cm >1.9 cm and buttock). The particle size distribution was then calculated as a percentage of each particle size class in relation to the whole mixed sample.

The bulk density of the willow chips was determined using fresh material from piles. The bulk density of the willow was determined using the mass per unit volume technique with a glass beaker, previously weighed [14]. Three measures were performed depending on the heterogeneity of the material sampled and the repeatability of the results obtained. The bulk density (**BD**) was determined for each sample of the compost variant and expressed in kg m$^{-3}$ according to Formula 1:

$$BD = \frac{\textbf{Fresh compost mass}}{\textbf{Beaker Volume}} \tag{1}$$

## 2.2 Preparation of willow extracts

Composted willow wooden chips from piles were taken separately, dried and ground in a mill with a diameter of 0.1 mm sieve. From each term prepared in four replicates 5 g of biomass material (dry weight) and soaked in 100 ml of distilled water. The composting biomass of the willow was kept for 24 hours in darkness, then the extract was filtered using Whatman filter paper no. 1 to remove fiber, lees, and other contamination. The sterilized distilled water was used as the control solution.

## 2.3 Germination experiments

Germination experiments were performed twice in 2020 and 2021 in 10 cm diameter Petri dishes. Each Petri dish before starting the experiment was sterilized at 105˚C for 4 hours and seeded with 50 seeds of the tested species *Sinapis alba* L.–white mustard. In the testing phase in both series, there were 96 dishes with the same concentration of willow water extract (5%) and 4 dishes with sterilized distilled water (control treatment). Petri dishes with seeds were placed into a vegetation chamber with 90% humidity. Germination experiments were carried out for seven days, maintaining a temperature regime of 22˚C for the constant temperature of the day (12 h) and 19˚C at night (12 h). After three days and the seventh day the germination index was performed. Three days after the beginning of the experiment in the vegetation chamber, 3 ml of the extract (and sterilized water in the control variant) were added to the Petri dishes. The experiment was conducted in 4 replicates.

The percentage of germination was calculated for each Petri dish separately (Eq 1). The length of the root (cm), the length of the cotyledons (cm), and the mass of the seedling (g) was measured for selected 10 seedlings per Petri dish. The fresh mass of the seedlings was determined for 10 seedlings to evaluate the allelopathic/toxicity potential of the extract. The experimental results comprised the following parameters:

The number of seeds germinated in each treatment; the percentage of germination in each variant (%). The germination seeds (GS%) were determined by Eq (2) [22].

$$\mathbf{GS}\ (\%) = \frac{\textbf{Number of seed germinated}}{\textbf{Total number of seed plated}} \times 100 \qquad (2)$$

Effect of inhibition according to the formula (Eq 3) for the II date of compost samples collection only:

$$\mathbf{IE} = \frac{\mathbf{C - T}}{\mathbf{C}} \cdot 100 \qquad (3)$$

C—measurement at the control variant for *Sinapis alba* L.

T–measurement at each treatment

In the conducted research, cultivated crops (willow and white mustard) were used. According to the national regulations, the use of these species for experimental purposes does not require any special permit. Our study complies with relevant institutional, national, and international guidelines and legislation.

## 2.4 Statistical analysis

Changes in nitrogen content, bulk density, and particle size distribution of the samples over and after composting were analysed using Fisher test with LSD to assess the effect of composting on different dates for willow chips variants. Differences between variants were determined by analysis of variance ANOVA MANOVA at the significance level of 0.05 or 0.01. In the variance analysis, in (four) replications, repeated in each variant of composting. The calculations (average) were performed with Statistica 13.1. The correlation coefficient (R) was determined with Microsoft Excel 2010. The correlation between total nitrogen and morphological traits of *Sinapis alba* L. has been performed by Pearson correlation.

## 3. Results and discussion

### 3.1. Temperature changes and moisture control

During the composting process changes of temperature were observed mainly in the composting piles with additives (WN, WF, WNF). The thermophilic phase was achieved in the mentioned variants 14 days after applying the additives (WN and WNF) and lasted no longer than 7 days.

In the control pile with only willow chips the thermophilic phase was not reached. The main reason for that was improper C:N ratio (over 50). According to Storey et al. [19], the increase in temperature during composting reduces the population of mesophilic bacteria, and the share of microflora in the further stage of composting depends on the availability of oxygen. Stentiford [23] found that increase temperature to 35-40°C favors increase of microbial diversity while higher increase to 45-55°C enhances the biodegradation rate. In our experiment only variants WN and WNF reached the microbial diversity phase (Fig 1). Negative correlation between temperature and oxygen concentration have been described by [24,25] who found that if the temperature of the composted mass was in the range of 65–70°C, the transformation processes were similar to the anaerobic conditions, regardless of the number and size of the air pores in the pile. In our study, the woodchips were 4–12 mm in size and the pile was highly aerated, which could also have affected the composting process.

In the experiment the initial material was characterized by low moisture content (26–35%) that resulted in low degradation rate of the composting material in the first ten days (Fig 2).

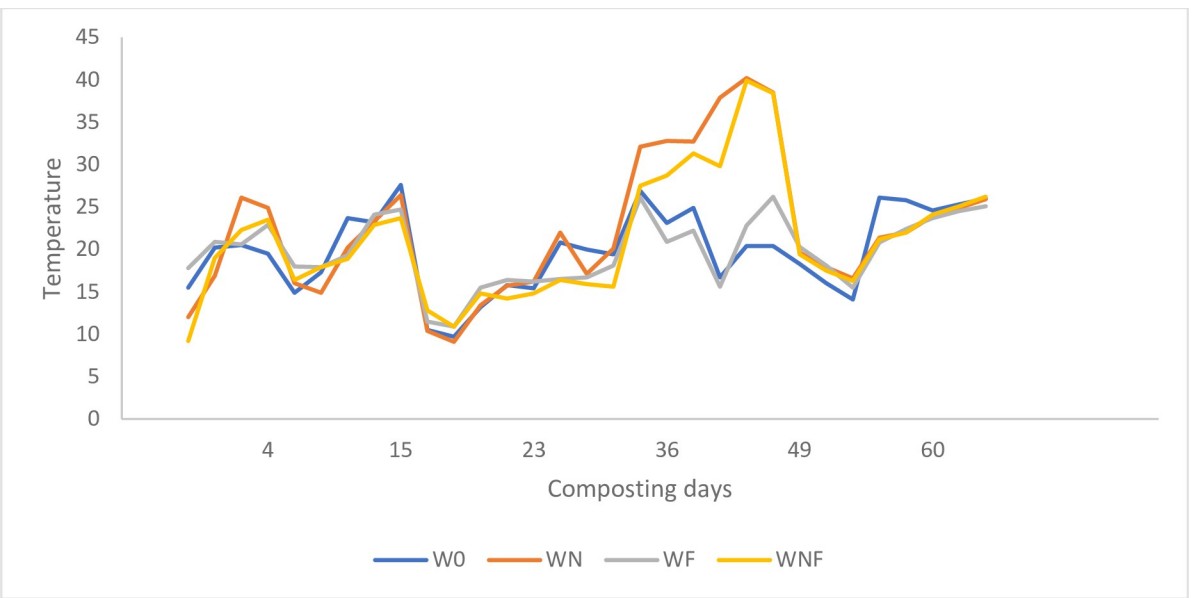

**Fig 1. Changes of temperature during composting of willow biomass.**

After 20 days of the process the piles were watered to the optimum level and the water content of the composted material was regularly replenished to maintain 40–60% $H_2O$ until the end of the experiment. The moisture content of the composted material is one of the most important technical parameters of the process. The optimum value should be contained within the range

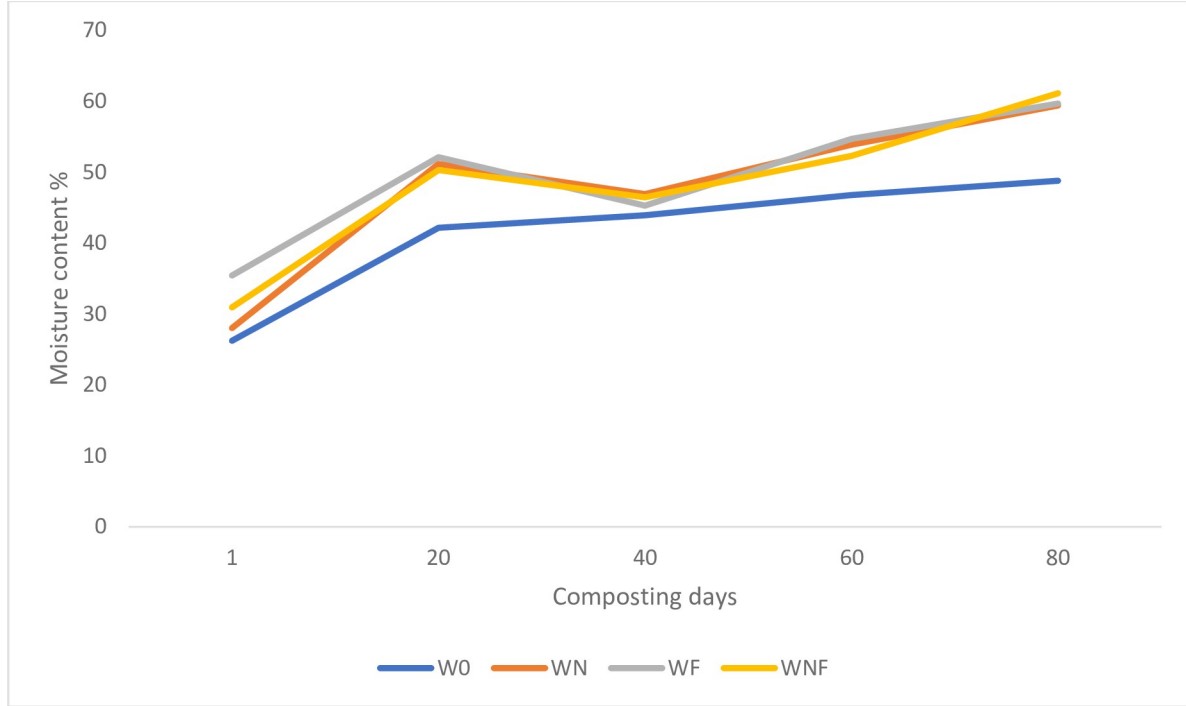

**Fig 2. Moisture content during composting of willow biomass.**

40–60% [26]. This parameter influences/stimulates the biodegradation rate, thermal, structural properties of the composted materials and affects microbial activity.

It must therefore be concluded that the composting of willow biomass using the semi-dynamic open pile method requires the control of moisture content and its immediate supplementation in the presence of a $H_2O$ deficiency.

## 3.2. Characteristics of willow composting biomass

During the 6-month composting process, the biomass sampled was characterized by varying nitrogen content (Fig 3). Throughout the sampling period, the N content was highest in the WN and WNF variant and ranged from 3.43 (at date VI) to 10.21% (one month after piles formation). The N content was similar when the willow was mixed with nitrogen and mycelium WNF (range from 3.43 to 8.48%). Willow biomass without nitrogen additives or mixed with mycelium had a similar nitrogen content throughout the composting period and ranged from 1.08 to 1.35% (W0) and from 1.18 to 1.53% (WF).

Especially the carbon to nitrogen ratio (C:N) is important in composting because microorganisms need a good balance of carbon and nitrogen (between 25 and 35) to be active. Both, too low and too high C:N ratio are inappropriate [27]. Ensuring an optimal C:N ratio allows one to gain balance between basic components used by the microorganisms during the composting process; however, not always possible to be fulfilled especially when materials that are added to the composted biomass, leads to the formation of substances having a negative impact on enzymatic activity of microorganisms [28]. Natural composting processes can change the carbon: nitrogen (C:N) ratio during process [29] leading to changes in the amount of particles less than 3.15 mm [30]. Sullivan et al. [31] recommend the use of nitrogen when

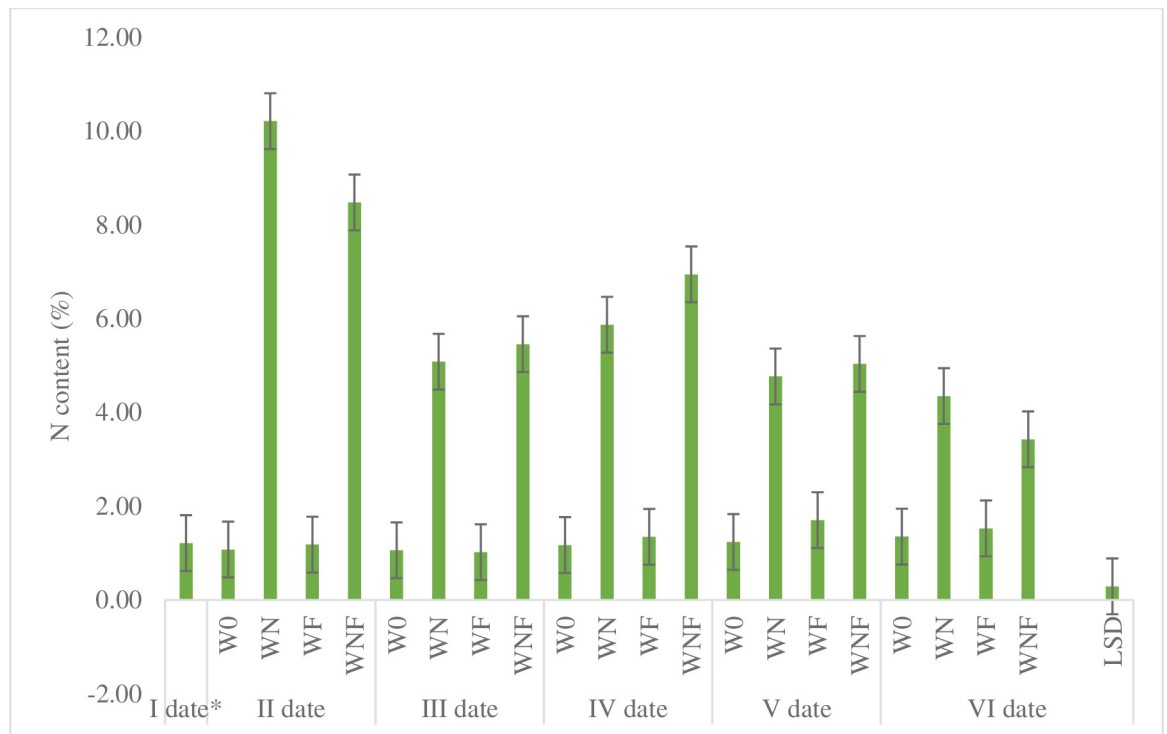

**Fig 3. The effect of composting processes on total N content (%) for different terms of sampling and variants of composting.**
*before composting, LSD Least Significant Difference.

woody biomass is composted and the ratio is greater than 20:1. Composting willow with the addition of nitrogen and mycelium contributed to a faster reduction in nitrogen content as a result of biomass decomposition and mineralization. In the WN and WNF variants, the addition of nitrogen accelerated the decomposition process of the willow biomass and the temperature in the compost heaps where nitrogen was used was higher between the 30th and 50th day of composting. During the initial phase of composting, nitrogen is generally not used by microorganisms at the same rate as carbon due to significant organic decomposition. As a result, the total nitrogen determined by Kjeldahla increases as a concentration effect, but in the study of Chan et al. [32] the increase was not significant similarly to our results where a higher N content in the compost was observed in the variants with nitrogen addition.

Modification of the composting process affected changes in the structure of the compost (Fig 4). The percentage of particle fractions in compost without additives in the second and third time periods was at a similar level. In the later period, the percentage of fractions >1.9 cm decreased, while the highest increase was observed in the percentage of fractions 0.8–1.9 cm. Similarly, a change in the structure of the biomass occurred after nitrogen application, and at the second date a decrease in the percentage of the largest fraction was observed, while the increase in the percentage of fractions 0.8–1.9 and <0.8 cm was at a similar level. The most

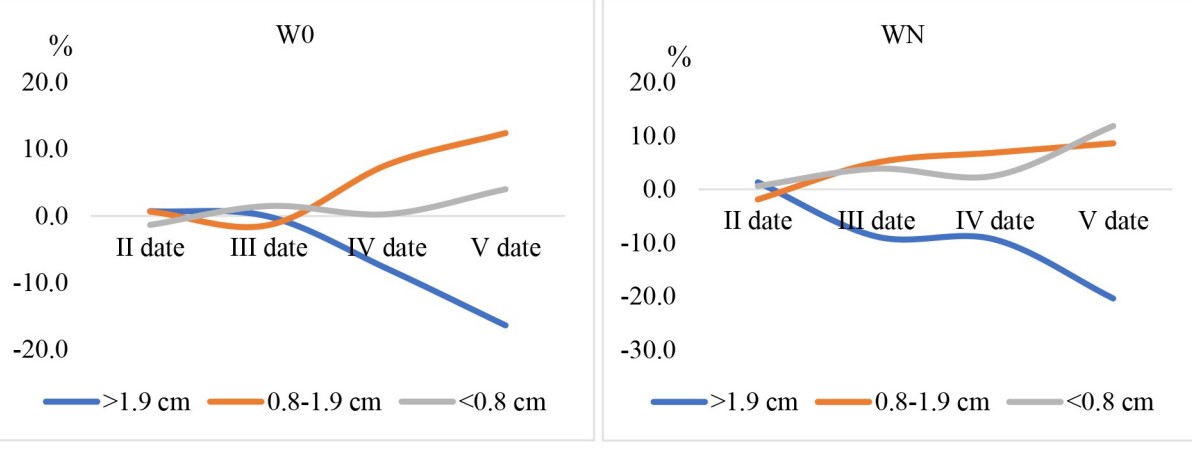

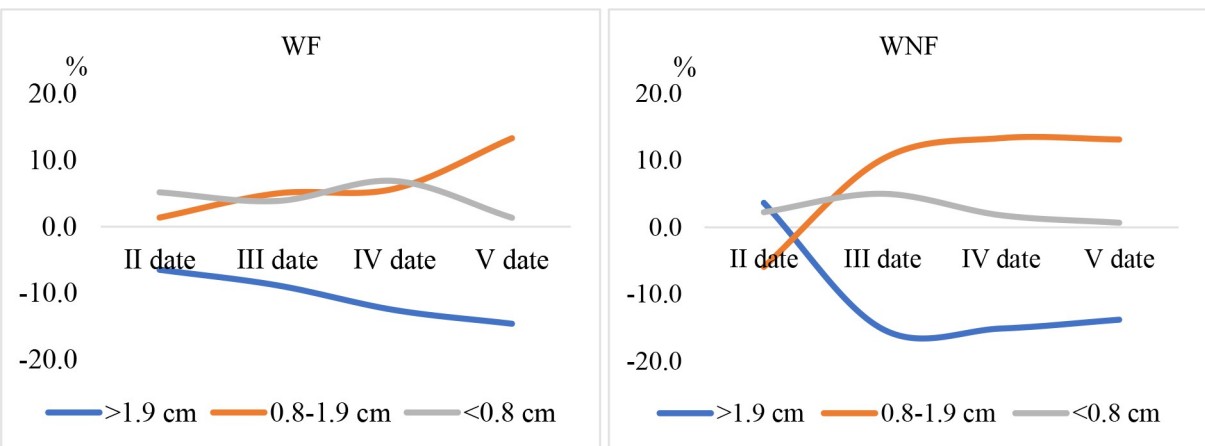

**Fig 4. The structure of the compost biomass changes over the composting process with various additives to the compost.** Percentage changes in the beginning of the biomass structure process (in different composting dates and variants).

dynamic changes in structure occurred after the application of fungi (WF) and the percentage of fractions >1.9 cm successively decreased, while the percentage of the other two fractions increased. At the end of the measurement period, the percentage of fractions <0.8 cm was at the level of the initial values. In the WNF variant, the change in biomass structure was similar to that of the WF, but a smaller increase in the percentage of fraction <0.8 cm was observed.

Chipped willow biomass has a particle structure that does not conform to the optimal parameters for a horticultural substrate. Peat, as a standard substrate used in horticulture, has a dominant fraction below 1 cm [33]. It was found that in the composting process, irrespective of treatment, influenced the reduction in the percentage of fraction >1.9 cm and in the chipped willow, nearly 60% was the fraction below 0.8 cm. The study by Whittaker et al. [29] also found noticeable changes in particle size during the composting period for woodchips. In woodchips, there was a 10% increase in the smaller fraction, 3.15–16 mm, after composting. In contrast, the number of small particles less than 3.15 mm, did not change significantly. In their research, that was related to a significant decrease in the bulk density. In our study, the addition of mycelium and nitrogen with mycelium affected the dynamics of these changes, which had already occurred in the initial stage of composting.

The bulk density of the compost depended on the changes that occurred during the composting. In the biomass of willow without additives (W0), there was a successive increase in bulk density from 179 (I date) to 297 (V date) kg m$^{-3}$ (Fig 5). Willow biomass with mycelium additions after an initially large increase in bulk density to 274 kg/m$^3$, the bulk density stabilized at 240 kg m$^{-3}$ on the V sampling date. After application of nitrogen and nitrogen with mycelium, the effect on bulk density was similar on all sampling dates and on the date V the weight of 1 m$^3$ of compost was at the level of 285 and 292 kg (WNF and WN, respectively).

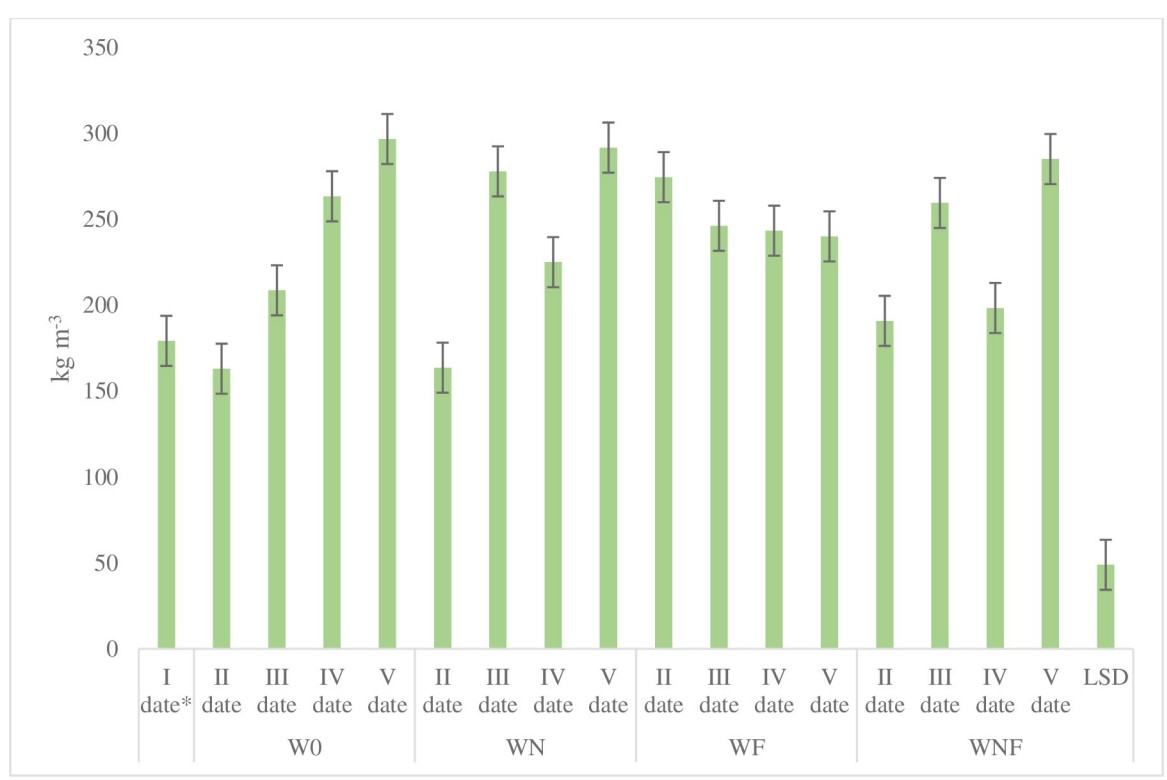

**Fig 5. Changes in bulk density depending on the treatment of willow composting and the sampling date (kg m$^{-3}$) (average for different terms and variants of composting).**

During the biomass biotransformation process, there was an increase in bulk weight from 34 (WF) to 66% (W0). The bulk density is the important element that influences the optimal conditions for the plant development, microbial activity, the degradation of organic matter, as well as the change in various mechanical properties, such as strength and porosity [34]. The bulk density ranged from 420 to 655 kg m$^{-3}$ for different types of composts [25]. According to Jain et al. [35] the bulk density was increased from 312 to 380 kg m$^{-3}$ and the volume reduction decreased significantly. Other studies conducted on the composting of various organic wastes have also shown a similar pattern for bulk density [36]. It is proved that bulk density of compost characterizes with decreasing tendency while the total organic matter of the compost increases. In our study, this parameter is changeable depending on the date of sampling and addition of nitrogen or mycelium.

Substrate bulk density is a very important indicator that determines water capacity, water holding capacity as well as labor input in the production process. Khoshand and Fall [33], report that the addition of sand ensures optimal water conditions, provided that its proportion does not exceed 36%. The bulk density of the substrate after addition of sand was within the range of 402–1004 kg m$^{-3}$ (peat 375.4 kg m$^{-3}$). Zoltai et al. [37] determined the bulk density for peat within the range of 150–200 kg m$^{-3}$. The composting process increased the bulk density compared to the initial material (179 kg m$^{-3}$). The highest increase (by 55%) was found in willow compost in which the process was not modified and at date VI it was 297 kg m$^{-3}$ and was in the estimated range for peat.

### 3.3 Bioassay of willow biomass during the composting process

On the basis of analyses conducted to evaluate the impact of willow compost with different additives, a differential response was found on the growth and development of the test seed of *Sinapis alba* L. (Photo 1). The aqueous extract of willow, prepared from biomass sampled before the start of the experiment, did not show a significant negative effect on the germination and morphological parameters of the white mustard seedlings (Photo 1, Table 2).

However, no emergence or a marked inhibition of emergence was observed in the willow biomass extract given to the composting process in the first period of the process. In date II, no germination of mustard seeds was observed in the WN variant extract and in the WNF variant the germination capacity was significantly lower. Similarly, significantly lower germination capacity was found for WNF in period IV. At the last sampling date (VI), regardless of the composting method, the germination capacity was statistically at the same level as that obtained under control conditions. The length of the embryonic roots, despite the large

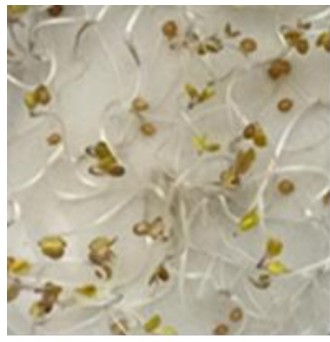 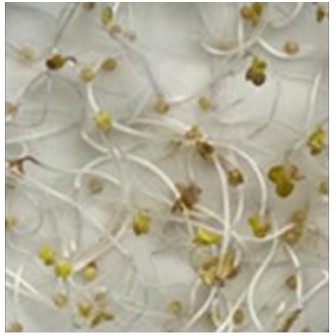 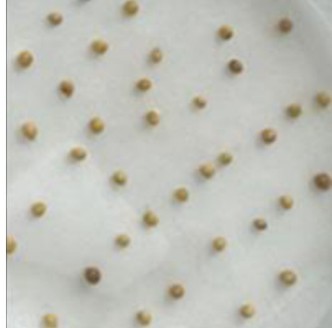

| Control water | II date W0 | II date WN |

**Photo 1. Germination of *Sinapis alba* L. seeds after 7 days of experiment established for the control and II date for W0 and WN treatment.**

**Table 2. Germination capacity and seedlings parameters depend on composting treatment and sampling date (average and standard deviation (±) for different terms of sampling and variants of composting).**

| Compost treatment | Date | Germination index (%) | Root length (cm) | Hypocotyl length (cm) | Seedling mass (g) |
|---|---|---|---|---|---|
| Control (water) | | 96.8±0.07 | 4.4±0.60 | 2.8±1.29 | 0.035±0.009 |
| I term before composting | | 93.0±0.09 | 2.4±0.42 | 2.8±0.22 | 0.031±0.008 |
| W0 | II | 95.1±0.18 | 4.8±1.26 | 3.5±0.10 | 0.038±0.011 |
| | III | 94.4±0.09 | 6.6±1.41 | 3.4±0.18 | 0.047±0.007 |
| | IV | 97.1±0.14 | 5.1±0.57 | 3.3±0.13 | 0.045±0.006 |
| | V | 95.7±0.06 | 6.3±0.53 | 3.2±0.20 | 0.046±0.008 |
| | VI | 97.6±0.06 | 8.1±1.84 | 3.4±0.10 | 0.048±0.004 |
| WN | II | 0.0±0 | 0.0±0 | 0.0±0 | 0.0±0 |
| | III | 91.1±0.09 | 2.0±0.60 | 2.1±0.19 | 0.034±0.005 |
| | IV | 92.5±0.06 | 0.8±0.51 | 0.9±0.12 | 0.019±0.007 |
| | V | 89.8±0.08 | 2.3±1.42 | 2.5±0.56 | 0.034±0.008 |
| | VI | 94.1±0.04 | 3.0±1.97 | 2.5±0.27 | 0.036±0.007 |
| WF | II | 97.6±0.07 | 5.5±0.71 | 3.5±0.32 | 0.043±0.005 |
| | III | 95.6±0.05 | 6.6±0.63 | 3.3±0.68 | 0.043±0.007 |
| | IV | 97.3±0.18 | 5.9±0.64 | 3.3±0.19 | 0.043±0.010 |
| | V | 95.0±0.06 | 5.9±085 | 3.5±0.34 | 0.047±0.004 |
| | VI | 98.2±0.14 | 7.1±1.68 | 3.2±0.37 | 0.048±0.009 |
| WNF | II | 49.0±0.08 | 0.1±0.17 | 0.2±0.43 | 0.002±0.004 |
| | III | 93.4±0.16 | 1.5±0.73 | 1.6±0.39 | 0.027±0.009 |
| | IV | 73.0±0.10 | 0.2±0.19 | 0.7±0.46 | 0.010±0.008 |
| | V | 90.1±0.07 | 2.2±1.21 | 1.6±0.43 | 0.031±0.007 |
| | VI | 97.3±0.11 | 3.3±0.19 | 3.0±0.43 | 0.044±0.012 |
| LSD (α = 0.05) | | 8.3 | n.s. | 1.2 | 0.007 |

differences, was statistically different. This was probably due to the large variation in rootlet length within the sample. The dependence of stimulation of embryonic root growth was found on the second sampling date in extracts from W0 and WF composted biomass. In the last sampling period (VI), they were respectively 84 and 61% longer than in the control. This relationship was not found on all sampling dates in the WN and WNF variants.

Inhibition of hypocotyl growth was found when extracts were prepared from biomass from WN and WNF. When nitrogen was added to the composted biomass, the toxic effect disappeared in period V, while the addition of nitrogen and mycelium to the composted willow inhibited the growth of hypocotyl until period VI. Composting willow without additives (W0) and with the addition of mycelium had no negative effect on hypocotyl growth and was 14–25% longer than seeds germinated in water.

Composting of willow with nitrogen addition (WN) as well as with nitrogen and mycelium addition (WNF) in the initial period had a negative effect on seedling weight. From date V onward, the seedling weight in the WN and WNF variants was statistically at the same level as in the control. An inhibitory effect of extracts on seedling weight was found in the earlier sampling periods. Willow compost and willow compost with added fungi had a positive effect on seedling weight and a weight increase of 9–40% was observed compared to the control variant. Willow bark extracts, contain many phytohormones, such as salicylic acid, which show a stimulating effect on root system growth, mitigation of environmental stresses, and flowering [38,39]. In our study, we found a stimulating effect of willow compost extracts and compared to seedlings obtained under control conditions in W0 and WF extracts, mustard seedlings at

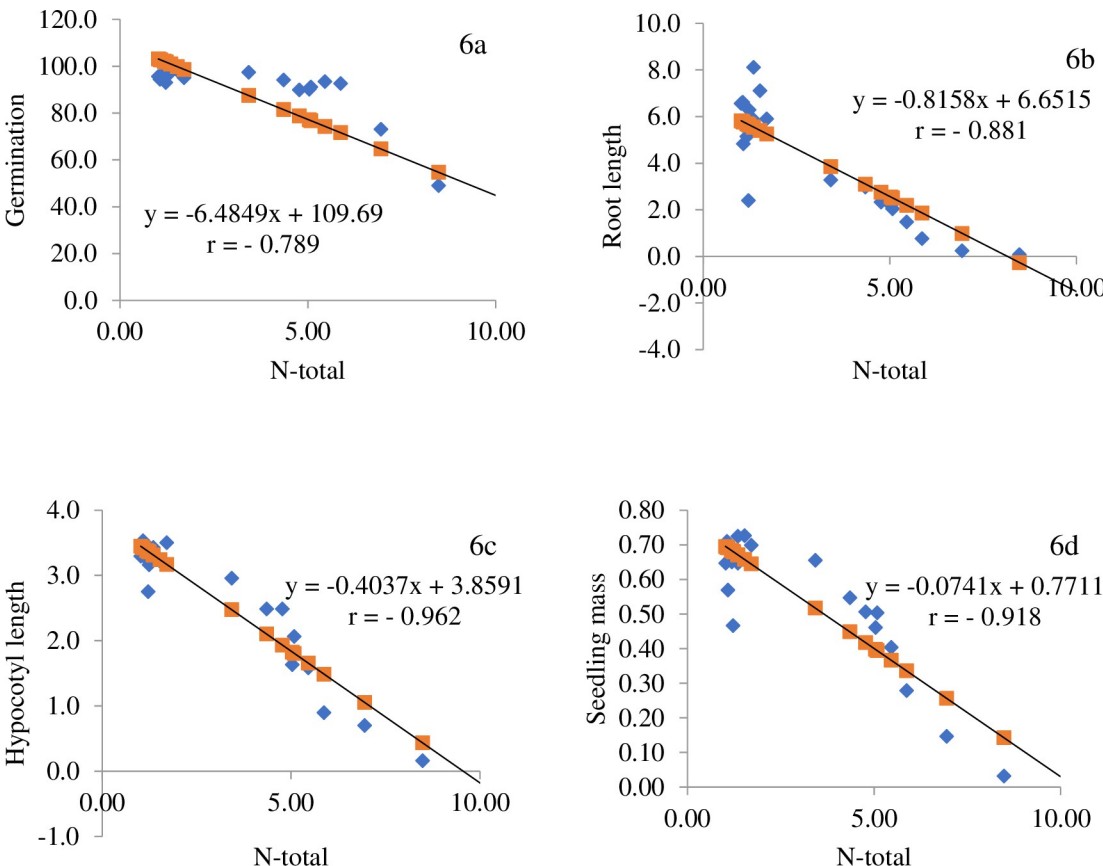

**Fig 6. Effect on total N content on the germination capacity and seedling parameters of the tested plant *Sinapis alba* L.**

all sampling dates had longer embryonic roots and hypocotyl. Such a tendency was not observed when extracts were prepared from WN and WNF compost.

Pearson correlation analysis showed a strong correlation between the germination capacity and parameters of the mustard seedlings and the nitrogen content of the composted biomass (Fig 6). Too high nitrogen content was toxic and inhibited germination and growth of embryonic roots, hypocotyl and negatively affected seedling weight. To the greatest extent, nitrogen content influenced hypocotyl length and mustard seedling weight (Fig 6C and 6D).

The difference in germination capacity of seeds in different media may be caused by different nutrient contents, as confirmed by Atiyeh et al. [40]. The degree of inhibition increased with the concentration of the extract. In our research inhibitory effect was observed while fungi addition to compost. Adamczewska-Sowińska et al. [9], Chen et al. [41] and Sun et al. [42], and report that excessive nitrogen availability can inhibit germination and initial plant growth. In a study by Zhang et al. [43], nitrogen content had a positive effect on the germination of eight plant species. In our study, the high nitrogen content was toxic to white mustard, and the seeds did not germinate, and the lower nitrogen content in the compost had a positive effect on the initial development of the test plant.

For the results obtained in period II, the inhibitory effect was calculated (Fig 7). Composting of willow biomass with nitrogen in the initial period had a toxic effect on the mustard germination process. A similar relationship was found when the composting process was supported by nitrogen and mycelium. The extract of composted willow without additives and

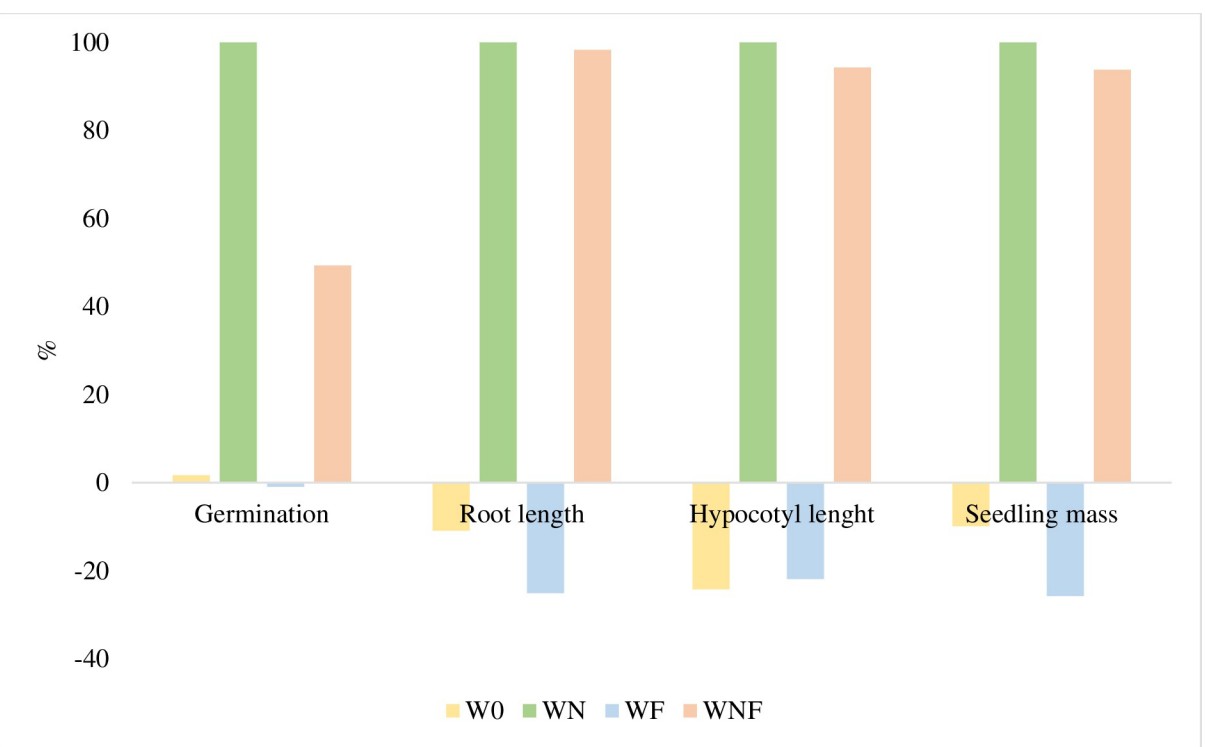

**Fig 7. Inhibition (+) and stimulation (-) effect of extract from willow composting biomass in percentage of seedling and germination parameters collected at the II term compared to the control (mustard seeds germinated in distilled water).**

with the addition of mycelium had a stimulating effect on the growth of mustard seedlings and compared to the control, the length of the embryonic roots, hypocotyl, and seedling mass were approximately 20% higher than in the control. Such a relationship was found in the WF compost extract. The beneficial effect on mustard seedlings of the W0 compost extract was smaller, in particular for embryonic root length as well as seedling weight (about 10% each).

Compost produced from willow, whose biotransformation process was aided by mycelium, had beneficial stimulating effects on seedling growth. Hayat and Ahmad [38], demonstrated the beneficial effect of salicylic acid on many plant growth processes. The stimulating effect of secondary metabolites found in willow can be inhibited by the high nitrogen content in the medium, especially immediately after application.

## 4. Conclusions

The production of willow compost must comply both with the process requirements arising from the appropriate C:N ratio and with the quality of the substrate produced. The addition of nitrogen, used due to its positive influence on the biomass biotransformation process, however can be toxic to plants. Therefore, it is necessary to determine the maximum dose that will ensure the correct course of the mineralization process and will not lead to nitrogen depletion of the composted biomass and inhibition of the process or excessive nitrogen accumulation. Composting woody biomass rich in cellulose and lignin's does not have such a dynamic progression, while the addition of nitrogen enhances its course and in the initial period the nitrogen content is too high for the correct development of the seedlings. With subsequent stages of composting, the nitrogen content decreased and the mature compost (date VI) showed no toxic effect on the test plant. The research showed that the addition of mycelium did not have

a toxic effect on the test plant, and the production effect and the beneficial effect on the change in the structure, as well as the biological evaluation confirmed the assumptions adopted. The study showed that in terms of quality, the willow compost fulfilled the quality requirements and the plant test showed in some variants a stimulating effect on the initial growth of white mustard.

## Acknowledgments

### Patents

Some of the results presented in this article were used in a patent filed on 28.06.2020 at the Polish Patent Office No. P.435103. A legal procedure is currently underway.

## Author Contributions

**Conceptualization:** Józef Sowiński, Elżbieta Jamroz.

**Data curation:** Anna Jama-Rodzeńska, Peliyagodage Chathura Dineth Perera, Jakub Bekier.

**Formal analysis:** Józef Sowiński, Peliyagodage Chathura Dineth Perera, Elżbieta Jamroz, Jakub Bekier.

**Investigation:** Anna Jama-Rodzeńska, Jakub Bekier.

**Methodology:** Józef Sowiński.

**Resources:** Józef Sowiński, Anna Jama-Rodzeńska, Peliyagodage Chathura Dineth Perera.

**Software:** Peliyagodage Chathura Dineth Perera, Elżbieta Jamroz.

**Validation:** Józef Sowiński.

**Writing – original draft:** Józef Sowiński, Anna Jama-Rodzeńska, Peliyagodage Chathura Dineth Perera, Elżbieta Jamroz, Jakub Bekier.

**Writing – review & editing:** Józef Sowiński, Elżbieta Jamroz.

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
