## [Decision Letter · Decision Letter 0]

16 Aug 2022

PONE-D-22-15059The changes of willow biomass characteristics during the composting process and their phytotoxicity effect on Sinapis alba L.PLOS ONE

Dear Dr. Sowiński,

Thank you for submitting your manuscript to PLOS ONE. After careful consideration, we feel that it has merit but does not fully meet PLOS ONE’s publication criteria as it currently stands. Therefore, we invite you to submit a revised version of the manuscript that addresses the points raised during the review process. Based on the reviewers' recommendation, the manuscript requires MAJOR REVISIONS before it can be considered for publication.

We look forward to receiving your revised manuscript.

Kind regards,

Nor Adilla Rashidi, Ph.D.

Academic Editor

PLOS ONE

Journal Requirements:

2. In your Methods section, please provide additional information regarding the permits you obtained for the work. Please ensure you have included the full name of the authority that approved the field site access and, if no permits were required, a brief statement explaining why

3. We noted in your submission details that a portion of your manuscript may have been presented or published elsewhere. [DETAILS AS NEEDED] Please clarify whether this [conference proceeding or publication] was peer-reviewed and formally published. If this work was previously peer-reviewed and published, in the cover letter please provide the reason that this work does not constitute dual publication and should be included in the current manuscript.

“The research is co-financed under the Leading Reasarch Groups support project from the subsidy increased for the period 2020–2025 in the amount of 2% of the subsidy referred to Art. 387 (3) of the Law of 20 July 2018 on Higher Education and Science, obtained in 2019.”

Please respond by return e-mail so that we can amend your financial disclosure and competing interests on your behalf.

Reviewers' comments:

Reviewer's Responses to Questions

**Comments to the Author**

1. Is the manuscript technically sound, and do the data support the conclusions?

Reviewer #1: Yes

Reviewer #2: Yes

Reviewer #3: Yes

2. Has the statistical analysis been performed appropriately and rigorously? 

Reviewer #1: Yes

Reviewer #2: Yes

Reviewer #3: Yes

3. Have the authors made all data underlying the findings in their manuscript fully available?

Reviewer #1: Yes

Reviewer #2: Yes

Reviewer #3: Yes

4. Is the manuscript presented in an intelligible fashion and written in standard English?

Reviewer #1: Yes

Reviewer #2: Yes

Reviewer #3: Yes

5. Review Comments to the Author

Reviewer #1: This paper presented a study on the properties of compost produced from willow chips. The particle size structure, bulk density and total nitrogen content were evaluated. The effectiveness of the compost was tested on Sinapis alba L. The morphology characteristic of the sample during germination was observed. Although considerable effort was invested into the planning and experimental stage of the study, the result of the study does not appear to be very promising. Moreover, critical evaluation on the performance of compost is needed to convince readers of the significance of the study. Readers should be able to find out the performance of the compost as a fertilizer/ growth stimulant from the abstract. However, this main point was lacking in the manuscript, especially in the abstract. The manuscript also needs to be proofread. Therefore, careful revision on the manuscript is needed to warrant a publication in PLOS ONE. Hope below comments will able to help to further improve the paper.

Specific comment:

Abstract:

- The background of the study is provided but the knowledge gap in the study seems to be missing.

- Also, significance of the study and how the findings can be used to advance the field should be included.

- An abstract is often presented separately from the article, so it must be able to stand alone.

- Kindly highlight the main findings that draw attention to the effectiveness of the compost.

- Please try to merge all information into a paragraph with some attractive and new findings. The main result from the review is not seem in the abstract.

- Kindly refer some latest papers as it is highly relevant to this report. Example, biological remediation of acid mine drainage: Review of past trends and current outlook; microalgae biorefinery: High value products perspectives; recent advances in downstream processing of microalgae lipid recovery for biofuel production

- How does this review fill in any knowledge gap?

Introduction:

- Line 41: correct problem to problems.

- Line 42: correct facing to faced by

- Line 45: correct way to ways.

- Line 46: What does the author mean by “in opposite increase”?

- Line 48: Please revise the sentence.

- Line 53: correct makes to make

- Line 58: Kindly break down the long sentence.

- Line 64: correct completely to complete

- Too many grammatical errors in this section. Please proofread.

- The introduction section is too lengthy and contain too much detail that can be omitted.

- Please revise section based on the structure below:

1st paragraph: Problem statement

2nd paragraph: Current ongoing solution

3rd paragraph: Proposed solution in this work.

4th paragraph: Summarized the current research novelty and objective of this work.

- There are some tips that improve structure from this paper that authors are recommended to refer: “Incorporating biowaste into circular bioeconomy/ A critical review of current trend and scaling up feasibility”.

- Problem statement of your introduction is not strong, need to discuss more about it.

- The earlier paragraphs should lead logically to specific objectives of the study.

- Note that this part of the Introduction gives specific details: for instance, the earlier part of the Introduction may mention the importance of this study whereas the concluding part will specify what methods of control were used and how they were evaluated.

Materials and methods:

- Line 153: The sentence “previously harvested in winter 2015” might be rephrased into: “The last harvest was in winter 2015”

- Line160: Please rephrase “this can regulate”.

- Since composting is one of the main focuses of the study, briefly explain the composting procedure.

- Provide the purity and origin of all the chemicals used.

- Include characterization study of the willow chips.

- State the number of replicates of the sample.

- Provide statistical analysis.

Results and discussion:

- Kindly update the references. For example, in line 274, the reference dated 1983 could be supported with latest study relevant to the field.

- Line 277: correct “to maximise” to “increase of”

- Kindly reformat all the figures according to guide for author. Normally, the background lines should be deleted.

- Line 283: Correct “what” to “that”

- Report the standard deviation of the values obtained.

- Line 315: Delete “being”

- Line 319: “an indicator…”. This sentence is confusing.

- How does the author benchmark the findings of current study to the literature?

- Explain disparity of C:N ratio in regulating composting process.

- The underlying mechanisms should be highlighted.

References

- Most of the references need to be updated.

Reviewer #2: 1. introduction: what the correlation between the GHGs emission and the utilization of lingnocellulosic biomass? Suggest the author supply this detail in this section.

2. introduction: the description in this section was lack of logic, so as to get the main point hardly. Suggest the author rewrite this section.

3. materials and methods: what the C/N ration of all samples did the author adjust in composting process?

Reviewer #3: This manuscript is about the use of willow biomass for composting process and its effect on Sinapis Alba L. Compost indexes, total nitrogen content, moisture content and other parameter studies have been conducted during composting of willow biomass. This study is interesting and important in ensuring the correct course of the mineralization process and C:N ratio. However, there are some limitations of this manuscript that need to be revised and confirmed by the authors as listed below:

1. Line 146: Are you using other indicator plants other than Sinapis Alba L since you also stated the use of Salix Viminalis L.

2. Line 163-166 and Line 177 182: It seems like a repetition of the definition of W0, WN, WF, and WNF.

3. Line 199: “…per unit volume technique (33)..”, what is “(33)”? Is it a citation number?

4. Line 257-267: Seem like the novelty aspect should be moved to the introduction section. I think this section more focusing on the methodology and procedure.

5. Line 293-294: “..low moisture content (26-35%) what resulted in low degradation rate of the composting material in the first ten days”, What do you mean by “that resulted in low degradation rate of the composting materials?

6. Line 294-297: Is this statement from your results or other studies since you cited a reference for this sentence?

7. Lien 303-304: From figure 3, the N content in WN variant is not always highest throughout the sampling period. For Iv and V date, the WNF variant is higher than WN.

8. I think you should highlight and discuss by focusing on your data/ results rather than the statement from other literature. It seems like I read it as a review paper. Most of the discussion from other papers, then 1-2 sentence/s explaining your result, then another discussion from other papers. It is hard to follow since most of the part focuses on the discussion from other papers rather than your results.

9. Line 481: “Adamczewska-Sowińska et al. (2021) Chen et al. (2020) Sun et al. (2018), and report that…” I think between Chen et al and Sun et al should have “and”.

I recommend minor corrections and the author needs to address carefully my concerns.

6. PLOS authors have the option to publish the peer review history of their article (what does this mean?). If published, this will include your full peer review and any attached files.

Reviewer #1: No

Reviewer #2: No

Reviewer #3: No

---

## [Author Response · Author response to Decision Letter 0]

2 Sep 2022

Thank you Reviewers for the comments and changing proposal of the manuscript entitled: "The changes of willow biomass characteristics during the composting process and their phytotoxicity effect on Sinapis alba L.".

All Reviewers proposal have been uploaded or comments point-by-point.

Reviewer #1: 

Question – Comments

This paper presented a study on the properties of compost produced from willow chips. The particle size structure, bulk density and total nitrogen content were evaluated. The effectiveness of the compost was tested on Sinapis alba L. The morphology characteristic of the sample during germination was observed. Although considerable effort was invested into the planning and experimental stage of the study, the result of the study does not appear to be very promising. Moreover, critical evaluation on the performance of compost is needed to convince readers of the significance of the study. Readers should be able to find out the performance of the compost as a fertilizer/ growth stimulant from the abstract. However, this main point was lacking in the manuscript, especially in the abstract. The manuscript also needs to be proofread. Therefore, careful revision on the manuscript is needed to warrant a publication in PLOS ONE. Hope below comments will able to help to further improve the paper.

Response

In the previous version, the authors specified what is the effect of willow compost on plants (line 132-141). 

In the revised version, the following sentence was added : "The beneficial effect of composted willow biomass as a horticultural substrate for growing tomato transplants and for growing lettuce at early development stages was confirmed in studies by Adamczewska Sowińska et al. 2021, Bekier et al. 2022."

The abstract was supplemented by adding :

Composted lignin-cellulosic biomass can replace peat and be used as a substrate for vegetable transplant production.

Specific comment:

Abstract:

Question – Comments

- The background of the study is provided but the knowledge gap in the study seems to be missing.

Response

Added

The impact of modifying the willow lignin-cellulosic biomass composting process has not been well analysed.

Question – Comments

- Also, significance of the study and how the findings can be used to advance the field should be included.

- An abstract is often presented separately from the article, so it must be able to stand alone.

- Kindly highlight the main findings that draw attention to the effectiveness of the compost.

- Please try to merge all information into a paragraph with some attractive and new findings. The main result from the review is not seem in the abstract.

- Kindly refer some latest papers as it is highly relevant to this report. Example, biological remediation of acid mine drainage: Review of past trends and current outlook; microalgae biorefinery: High value products perspectives; recent advances in downstream processing of microalgae lipid recovery for biofuel production

- How does this review fill in any knowledge gap?

Response

The literature data cannot be referred to in the abstract. As suggested by the Reviewer, the abstract has been supplemented with the information on what benefits result from obtaining good quality compost. As suggested by the Reviewer, the following sentences were included : "Peatlands and peat are of very important economic but above all environmental significance. The conservation of peatland resources is one of the most crucial future challenges. Composts and other forms of lignin-cellulosic biomass are potentially the best renewable alternative to peat in its economic use." 

The introduction of the manuscript elaborates on this issue in more detail.

Introduction:

Question – Comments

- Line 41: correct problem to problems.

Response

Corrected

Question – Comments

- Line 42: correct facing to faced by

Response

Corrected

Question – Comments

- Line 45: correct way to ways.

Response

Corrected

Question – Comments

- Line 46: What does the author mean by “in opposite increase”?

Response

This phrase has been removed.

Question – Comments

- Line 48: Please revise the sentence.

Response

The sentence has been revised: 

Peatland is one of most globally important stored carbon resources and the bound carbon stocks (in terms of quantity on a global scale) exceed those bound by tropical forests (500-700 billion tons of C compared 360 billion tons of C, respectively) [2].

Question – Comments

- Line 53: correct makes to make

Response

Corrected

Question – Comments

- Line 58: Kindly break down the long sentence.

Response

The sentence has been revised

Peatland drainage leads to degradation and, in conjunction with climate warming, rising temperatures, and drier weather patterns, which can lead to excessive mineralization and destabilization of peat C stores. In extreme cases fires may break out [2, 6]. Therefore its use is limited (also because of its long-lasting effect and reduced nutrient delivery to crops).

Question – Comments

- Line 64: correct completely to complete

Response

Corrected

Question – Comments

- Too many grammatical errors in this section. Please proofread.

Response

Corrected

Question – Comments

- The introduction section is too lengthy and contain too much detail that can be omitted.

- Please revise section based on the structure below:

1st paragraph: Problem statement

2nd paragraph: Current ongoing solution

3rd paragraph: Proposed solution in this work.

4th paragraph: Summarized the current research novelty and objective of this work.

Response

Introduction has been shortened by ¼ and less important elements have been removed. The chapter layout is in accordance with the Reviewer's comments. Section 2.5 has been removed and the content contained in the Novelty aspect has been placed at the end of the chapter Introduction.

Question – Comments

- There are some tips that improve structure from this paper that authors are recommended to refer: “Incorporating biowaste into circular bioeconomy/ A critical review of current trend and scaling up feasibility”.

Response

Thank you for the suggestion. The specification have been included in the text.

Question – Comments

- Problem statement of your introduction is not strong, need to discuss more about it.

Response

The basis for the study was the environmental and economic importance of peat and peatlands. The authors pointed out the need to protect them, due to their global strategic role in water retention as well as CO2 emissions resulting from their exploitation. Attention was drawn to the adverse processes resulting from peatland desiccation leading to their degradation e.g. by fires.

Question – Comments

- The earlier paragraphs should lead logically to specific objectives of the study.

Response

With the suggested amendments included, the Introduction now has a logical layout.

Question – Comments

- Note that this part of the Introduction gives specific details: for instance, the earlier part of the Introduction may mention the importance of this study whereas the concluding part will specify what methods of control were used and how they were evaluated.

Response

The Reviewer's comments addressed provide an appropriate (in the authors' view) character to the Introduction chapter.

Materials and methods:

Question – Comments

- Line 153: The sentence “previously harvested in winter 2015” might be rephrased into: “The last harvest was in winter 2015”

Response

Changed

Question – Comments

- Line160: Please rephrase “this can regulate”.

Response

Changed

Question – Comments

- Since composting is one of the main focuses of the study, briefly explain the composting procedure.

Response

Added:

The willow chip compost prisms were formed on a horticultural mat. The prisms were 1.5 m wide, 1.3 m high and 5 m long. The volume of the prisms was approximately 5 m3. The prisms were successively irrigated and manually mixed at 4-5 week intervals.

Question – Comments

- Provide the purity and origin of all the chemicals used.

Response

Reagents used for the determination of total nitrogen were as follows: sulphuric acid 96% (pure for analysis) - CAS catalogue number: 7664-93-9, granulated sodium hydroxide (at a concentration of 30%, pure for analysis) - CAS catalogue number: 1310-73-2, hydrogen peroxide 30% (pure for analysis) - CAS catalogue number: 7722-84-1.The reagents originally came from the company: P.P.H. "STANLAB" Sp. z.o.o Lublin.

Question – Comments

- Include characterization study of the willow chips.

Response

Added

The chops size ranged from 4 to 12 mm.

Question – Comments

- State the number of replicates of the sample.

Response

Added

The experiment was conducted in 4 replicates.

Question – Comments

- Provide statistical analysis.

Response

Added

Analyses were performed using the ANOVA MANOVA method.

Results and discussion:

Question – Comments

- Kindly update the references. For example, in line 274, the reference dated 1983 could be supported with latest study relevant to the field.

Response

Changed

Question – Comments

- Line 277: correct “to maximise” to “increase of”

Response

Changed

Question – Comments

- Kindly reformat all the figures according to guide for author. Normally, the background lines should be deleted.

Response

Changed

Question – Comments

- Line 293: Correct “what” to “that”

Response

Changed

Question – Comments

- Report the standard deviation of the values obtained.

Response

Changed

Question – Comments

- Line 315: Delete “being”

Response

Changed

Question – Comments

- Line 319: “an indicator…”. This sentence is confusing.

Response

Changed

Both, too low and too high C:N ratio are inappropriate.

Question – Comments

- How does the author benchmark the findings of current study to the literature?

- Explain disparity of C:N ratio in regulating composting process.

- The underlying mechanisms should be highlighted.

Response

In Chapter 3 Results and Discussion, the results of our own research were presented and compared to available literature data. The scope of the discussion was the variation in the composting process, the quality of the compost depending on the solutions used. An important part of the discussion was the optimum C:N ratio and the possibility of changing it through additives. The biological evaluation of the compost was particularly important and special attention was paid to it. The discussion highlighted differences in compost quality, the effect of the composting process on levels of nitrogen and on mustard seed germination.

References

- Most of the references need to be updated.

Reviewer #2: 

Question – Comments

1. introduction: what the correlation between the GHGs emission and the utilization of lingnocellulosic biomass? Suggest the author supply this detail in this section.

Response

In the reviewed version of the manuscript the issue of biomass bioconversion and emissions depending on the C:N ratio has been described in the section from line 106 to 126. Narrow C:N ratio accelerates organic matter decomposition process and increases GHG emissions. Based on the literature data, the most optimal C:N ratio is 30- 40.

Question – Comments

2. introduction: the description in this section was lack of logic, so as to get the main point hardly. Suggest the author rewrite this section.

Response

This section has been revised according Reviewer's suggestion. Section 2.5 has been removed and the content of the Novelty aspect section has been moved to the Introduction. We have revised this section according the following paragraphs:

1st paragraph: Problem statement

2nd paragraph: Current ongoing solution

3rd paragraph: Proposed solution in this work.

4th paragraph: Summarized the current research novelty and objective of this work.

Question – Comments

3. materials and methods: what the C/N ration of all samples did the author adjust in composting process?

Response

Raw willow chips are characterized by very wide C:N ratio - about 100 [6]. In the experiment nitrogen was added to reduce the C:N ratio to about 30, otherwise the composting process could not start.

Reviewer #3: 

This manuscript is about the use of willow biomass for composting process and its effect on Sinapis Alba L. Compost indexes, total nitrogen content, moisture content and other parameter studies have been conducted during composting of willow biomass. This study is interesting and important in ensuring the correct course of the mineralization process and C:N ratio. However, there are some limitations of this manuscript that need to be revised and confirmed by the authors as listed below:

Question – Comments

1. Line 146: Are you using other indicator plants other than Sinapis Alba L since you also stated the use of Salix Viminalis L.

Response

No. Salix viminalis biomass composted and water extract from composted biomass were used for the assessment of Sinapis alba seeds germination.

Question – Comments

2. Line 163-166 and Line 177 182: It seems like a repetition of the definition of W0, WN, WF, and WNF.

Response

Thank you. The information contained in lines 163-166 has been removed.

Question – Comments

3. Line 199: “…per unit volume technique (33)..”, what is “(33)”? Is it a citation number?

Response

Yes it was an item number in the literature list.

Question – Comments

4. Line 257-267: Seem like the novelty aspect should be moved to the introduction section. I think this section more focusing on the methodology and procedure.

Response

The proposal has been included. Reviewer 1 was of a similar opinion.

Question – Comments

5. Line 293-294: “..low moisture content (26-35%) what resulted in low degradation rate of the composting material in the first ten days”, What do you mean by “that resulted in low degradation rate of the composting materials?

Response

The willow biomass was chipped before composting. The size of the chippings was between 4-12 mm and the prism was very easily ventilated and therefore tended to dry quickly. After the initial period, the frequency of moisture control was increased and also the prisms were covered with transparent perforated film.

Question – Comments

6. Line 294-297: Is this statement from your results or other studies since you cited a reference for this sentence?

Response

Yes these are the results from our own research. The quoted item indicates that the moisture level was brought to an optimum level. The sentence was changed and the quoted literature item was removed.

Question – Comments

7. Lien 303-304: From figure 3, the N content in WN variant is not always highest throughout the sampling period. For Iv and V date, the WNF variant is higher than WN.

Response

Thank you. Corrected

Question – Comments

8. I think you should highlight and discuss by focusing on your data/ results rather than the statement from other literature. It seems like I read it as a review paper. Most of the discussion from other papers, then 1-2 sentence/s explaining your result, then another discussion from other papers. It is hard to follow since most of the part focuses on the discussion from other papers rather than your results.

Question – Comments

Response

To ensure the correct manuscript proportions, information from the study was added and information from other studies was removed.

In our study, the woodchips were 4-12 mm in size and the pile was highly aerated, which could also have affected the composting process.

In the WN and WNF variants, the addition of nitrogen accelerated the decomposition process of the willow biomass and the temperature in the compost heaps where nitrogen was used was higher between the 30th and 50th day of composting.

To the greatest extent, nitrogen content influenced hypocotyl length and mustard seedling weight (Figure 6c-6d).

The beneficial effect on mustard seedlings of the W0 compost extract was smaller, in particular for embryonic root length as well as seedling weight (about 10% each).

9. Line 481: “Adamczewska-Sowińska et al. (2021) Chen et al. (2020) Sun et al. (2018), and report that…” I think between Chen et al and Sun et al should have “and”.

Response

Corrected

I recommend minor corrections and the author needs to address carefully my concerns.

---

## [Decision Letter · Decision Letter 1]

13 Sep 2022

The changes of willow biomass characteristics during the composting process and their phytotoxicity effect on Sinapis alba L.

PONE-D-22-15059R1

Dear Dr. Sowiński,

We’re pleased to inform you that your manuscript has been judged scientifically suitable for publication and will be formally accepted for publication once it meets all outstanding technical requirements.

Kind regards,

Nor Adilla Rashidi, Ph.D.

Academic Editor

PLOS ONE

Additional Editor Comments (optional):

Reviewers' comments:

Reviewer's Responses to Questions

**Comments to the Author**

1. If the authors have adequately addressed your comments raised in a previous round of review and you feel that this manuscript is now acceptable for publication, you may indicate that here to bypass the “Comments to the Author” section, enter your conflict of interest statement in the “Confidential to Editor” section, and submit your "Accept" recommendation.

Reviewer #1: All comments have been addressed

Reviewer #2: All comments have been addressed

Reviewer #3: All comments have been addressed

2. Is the manuscript technically sound, and do the data support the conclusions?

Reviewer #1: Yes

Reviewer #2: Yes

Reviewer #3: Yes

3. Has the statistical analysis been performed appropriately and rigorously? 

Reviewer #1: Yes

Reviewer #2: Yes

Reviewer #3: Yes

4. Have the authors made all data underlying the findings in their manuscript fully available?

Reviewer #1: Yes

Reviewer #2: Yes

Reviewer #3: Yes

5. Is the manuscript presented in an intelligible fashion and written in standard English?

Reviewer #1: Yes

Reviewer #2: Yes

Reviewer #3: Yes

6. Review Comments to the Author

Reviewer #1: (No Response)

Reviewer #2: The authors have revised their manuscript according to the comments, in this case, it should be accepted for publication by PLOS ONE.

Reviewer #3: Thank you for addressing all my previous concerns carefully. Thus, I recommend for publication in PLOS ONE in the present form.

7. PLOS authors have the option to publish the peer review history of their article (what does this mean?). If published, this will include your full peer review and any attached files.

Reviewer #1: No

Reviewer #2: **Yes: **Zengqiang Zhang

Reviewer #3: No

---

## [Editor Report · Acceptance letter]

19 Sep 2022

PONE-D-22-15059R1 

The changes of willow biomass characteristics during the composting process and their phytotoxicity effect on *Sinapis alba* L. 

Dear Dr. Sowiński:

I'm pleased to inform you that your manuscript has been deemed suitable for publication in PLOS ONE. Congratulations! Your manuscript is now with our production department. 

Kind regards, 

on behalf of

Dr. Nor Adilla Rashidi 

Academic Editor

PLOS ONE